# Frequency Pooling: Shift-equivalent and Anti-Aliasing Down Sampling

## Abstract

Convolutional layer utilizes the shift-equivalent prior of images which makes it a great success for image processing. However, commonly used down sampling methods in convolutional neural networks (CNNs), such as max-pooling, average-pooling, and stride convolution, are not shift-equivalent. This destroys the shift-equivalent property of CNNs and degrades their performance. In this paper, we provide a formal definition of shift-equivalent when down sampling involved. We propose a *strict shift equivalent and anti-aliasing* pooling method. This is achieved by (inverse) Discrete Fourier Transform and we call our method frequency pooling. Experiments on image classifications show that frequency pooling improves accuracy and robustness w.r.t shifts of CNNs.

## 1 Introduction

Convolutional neural networks (CNNs) have achieved great success on image processing Goodfellow et al. (2016), natural language processing Yin et al. (2017), game playing Mnih et al. (2013) and so on. One of the reasons is that convolutions utilize the shift-equivalent prior of signals. Modern CNNs include not only convolutional layers but also down sampling or pooling layers. As an important part of CNNs, pooling layers are used to reduce spatial resolution of feature maps, aggregate spatial information, and reduce computational and memory cost.

Based on classical Nyquist sampling theorem Nyquist (1928), the sampling rate must be at least as twice as the highest frequency of a signal. Otherwise, frequency aliasing will appear, i.e. high-frequencies of the signal alias into low-frequencies. This leads sub-optimal when reconstruction the signals and misleads the following processing because orthogonal components are mixed again. To anti-alias, a traditional solution is that applying low-pass filter to the signal before down sampling it. Following the spirit of blurred down sampling, early CNNs use average pooling to achieve down sampling Lecun et al. (1989). Later, empirical evidence suggests max-pooling Scherer et al. (2010) and stride convolutions Long et al. (2015) provide better performance. They are widely used in CNNs but they don't consider about anti-aliasing.

Shift-equivalent is another expected property of pooling. Otherwise it will destroy the shift-equivalent of CNNs and thus the shift-equivalent prior of signals is not fully utilized. Unfortunately, most commonly used poolings are not shift-equivalent. Worse, small shifts in the input can drastically change the output when stacking multiple max-pooling or stride convolutions Engstrom et al. (2017); Azulay & Weiss (2018); Zhang (2019).

Shift-equivalent is believed to be a fundamental property of CNNs. However, the fact that CNNs with poolings are not shift-equivalent has been ignored by the community. Until recently, Zhang (2019) propose anti-aliasing pooling (AA-pooling) by low-pass filtering before down sampling. They observe increased accuracy and better generalization on image classification when low-pass filtering is integrated correctly. Specifically, they decompose a pooling operator with down sampling factor $s$ into two parts: a pooling operator with factor 1 and a blur filter with factor $s$. Although AA-pooling reduces the aliasing effects and makes the outputs more stable w.r.t input shifts, it is neither strict shift-equivalent nor anti-aliasing in theory.

In this paper, we propose a *strict shift equivalent and anti-aliasing* pooling in theory. We first transform a signal or image into frequency domain via Discrete Fourier Transform (DFT). Then we only retain its low frequencies, i.e. the frequencies which are smaller than Nyquist sampling

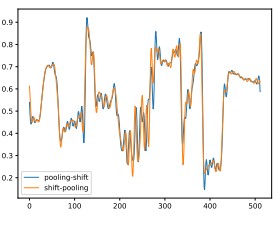
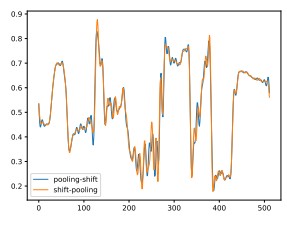
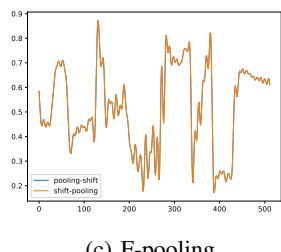

(a) Max-pooling        (b) Avg-pooling        (c) F-pooling

Figure 1: Simple tests of shift-equivalent. Blue lines are obtained by pooling, up sampling, and shift in order. Orange lines are obtained by shift, pooling, and up sampling in order. All Pooling operations down sample original signals by a factor 4. The shift operation shifts original signals by 2 pixels. The up sampling operation is set to equation 8. Best viewed on screen.

rate. Finally, we transform the low frequencies back into time domain via inverse DFT. We call our method frequency pooling (F-pooling). Note that a similar method is proposed in Ryu et al. (2018). However, they only focus on classification accuracy. Shift equivalent of their method is not evaluated in both theory and practice. Compared with previous works, the novelties and contributions of F-pooling are summarized as follows:

- We formally define shift-equivalent of functions which contain down sampling operations. A suitable up sampling operation $\mathcal{U}$ must be involved in the definition. Without it, the definition for discrete signal is ill-posed. We believe a formal mathematical treatment has great value for further research.

- We prove that F-pooling is the optimal anti-aliasing down sampling operation from the perspective of reconstruction. We also prove that F-pooling is shift equivalent. The up sampling operation $\mathcal{U}$ plays an important rule in our proofs. To best of our knowledge, F-pooling is the first pooling method which has those properties.

- Experiments on CIFAR-100 and a subset of ImageNet demonstrate that F-pooling remarkably increases accuracy and robustness w.r.t shifts of commonly-used network architectures. Moreover, the shift consistency of F-pooling is improved more when we replace zero padding of convolutions with circular padding.

## 2 METHOD

In this section, we first define shift-equivalent for CNNs formally. Then we describe F-pooling in detail and prove its properties. Finally we discuss our implementation and some practical issues.

### 2.1 DEFINITION OF SHIFT-EQUIVALENT

Denote $\mathbf{X}$ as an $h_0 \times w_0 \times c$ tensor where $h_0$, $w_0$, and $c$ are the height, width, and number of channels of $\mathbf{X}$ respectively. Denote $\mathbf{Y}$ as an $h_1 \times w_1 \times c$ tensor. We suppose $h_0 > h_1$ and $w_0 > w_1$. $\mathcal{M} : \mathbf{X} \rightarrow \mathbf{Y}$ is a pooling operation and $\mathcal{U} : \mathbf{Y} \rightarrow \mathbf{X}$ is a up sampling operation. We say $\mathcal{M}$ is shift-equivalent if and only if

$$Shift_{\triangle h, \triangle w}(\mathcal{U}\mathcal{M}\mathbf{X}) = \mathcal{U}\mathcal{M}Shift_{\triangle h, \triangle w}(\mathbf{X}) \tag{1}$$
$$\forall(\triangle h, \triangle w), \quad \exists \mathcal{U}$$

That is, if there is a suitable up sampling operation $\mathcal{U}$ which makes $\mathcal{U}\mathcal{M}$ and $Shift_{\triangle h, \triangle w}$ commutable, then $\mathcal{M}$ is shift-equivalent. The shift operation is required to be circular or periodic. When a shifted element hits the edge, it rolls to other side.

$$Shift_{\triangle h, \triangle w}(\mathbf{X}_{a,b}) = \mathbf{X}_{(a+\triangle h)\%h_0, (b+\triangle w)\%w_0} \tag{2}$$

where % is the modulus function. Our definition of shift-equivalent is similar with the one in Zhang (2019). The difference is that we introduce an up sampling $\mathcal{U}$ to make this definition strict. *Without*

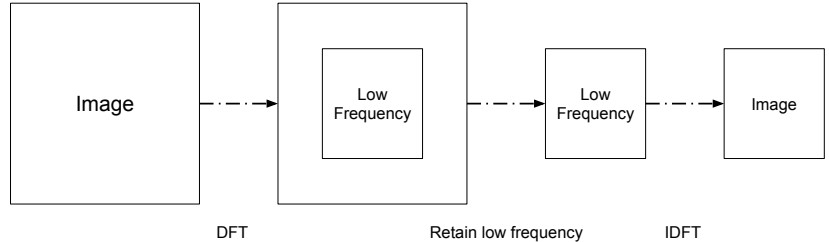

Figure 2: An illustration of the forward process of F-pooling. We assume the lowest frequencies are at center.

*introducing $\mathcal{U}$, the definition of shift-equivalent for pooling is ill-posed.* Suppose $\mathcal{M}$ down samples signals by a factor $s$. Without introducing $\mathcal{U}$, one may define shift-equivalent as follows:

$$Shift_{\triangle h/s, \triangle w/s} \cdot \mathcal{M}(\mathbf{X}) = \mathcal{M} \cdot Shift_{\triangle h, \triangle w}(\mathbf{X}), \qquad \forall (\triangle h, \triangle w) \tag{3}$$

However, $Shift_{\triangle h/s, \triangle w/s}$ is not operable for discrete signals when $\triangle h \% s \neq 0$ or $\triangle h \% s \neq 0$. To make the shifted element which is not on lattice operable, one must interpolate or up sample the down sampled signals. This leads to the definition in equation 1.

As in equation 1, to make $\mathcal{M}$ shift-equivalent, one must find the corresponding up sampling operation $\mathcal{U}$. For a given $\mathcal{M}$, the first line in equation 1 may hold for some up sampling operations. But it may not hold for other up sampling operations.

## 2.2 F-POOLING

The basic idea of F-pooling is removing the high frequencies of signals and reconstructing the signals only using the low frequencies. In this paper, *high frequencies mean the frequencies which are beyond Nyquist frequency*, i.e. half of signal resolution. And *low frequencies mean the frequencies which are not higher than Nyquist frequency*. To remove high frequencies, we first transform signals into frequency domain via DFT. Then only the low frequencies are retained. Finally, we transform the low frequencies back to time domain via inverse DFT (IDFT). Figure 2 gives an illustration of the forward process of F-pooling.

Since 2D (inverse) DFT can be decomposed into two 1D (inverse) DFT, we provide the formal representation of F-pooling in 1D case. Denote $\mathbf{x} \in \mathbb{R}^N$ as time domain signal and $\mathbf{y} \in \mathbb{C}^M$ as frequency domain signal. Without loss of generality, we suppose $M$ is even. $\mathbf{F}_N \in \mathbb{C}^{N \times N}$ is the so-called DFT-matrix:

$$\mathbf{F}_N = \begin{bmatrix} \omega_N^{0 \cdot 0} & \omega_N^{0 \cdot 1} & \dots & \omega_N^{0 \cdot (N-1)} \\ \omega_N^{1 \cdot 0} & \omega_N^{1 \cdot 1} & \dots & \omega_N^{1 \cdot (N-1)} \\ \vdots & \vdots & \ddots & \vdots \\ \omega_N^{(N-1) \cdot 0} & \omega_N^{(N-1) \cdot 1} & \dots & \omega_N^{(N-1) \cdot (N-1)} \end{bmatrix} \tag{4}$$

where $\omega_N = e^{-2\pi i/N}$. $\mathbf{F}_N^*$ is the inverse DFT-matrix. By definition, $\frac{1}{N} \mathbf{F}_N \mathbf{F}_N^* = \mathbf{I}$ where $\mathbf{I}$ is identity matrix. We denote $\mathbf{T}_\mu$ as a operation which selects the first $\mu$ rows and the last $\mu$ rows of a matrix.

$$\mathbf{T}_\mu(\mathbf{x}) = [\mathbf{x}_{\mathbf{1}:\mu}; \mathbf{x}_{N-\mu+1:N}] \tag{5}$$

That is we select the basis of frequencies ranged in $[-\mu, \mu)$, due to periodicity of DFT. When applying $\mathbf{T}_\mu$ to a signal, we obtain its lowest $\mu$ frequencies. Then the function of F-pooling for 1D signals is represented as:

$$\mathbf{y} = \frac{1}{N} \mathbf{F}_\mathbf{M}^* \mathbf{T}_{\frac{M}{2}} \mathbf{F}_N \mathbf{x} \overset{def}{=} \mathbf{P}\mathbf{x} \tag{6}$$

where $\mathbf{P} \in \mathbb{C}^{M \times N}$ is the transform matrix of F-pooling.

Now we formalize F-pooling for CNNs. Recall that $\mathbf{X}$ is an $h_0 \times w_0 \times c$ tensor and $\mathbf{Y}$ is an $h_1 \times w_1 \times c$ tensor. We apply F-pooling to each channel of $\mathbf{X}$.

$$\mathbf{Y}_{:,:,i} = \mathbf{P}_h \mathbf{X}_{:,:,i} \mathbf{P}_w^*, \quad i \in [1, c] \tag{7}$$

where $\mathbf{P}_h \in \mathbb{C}^{h_1 \times h_0}$ and $\mathbf{P}_w \in \mathbb{C}^{w_1 \times w_0}$ are the transform matrices of F-pooling along vertical direction and horizontal direction respectively. Since F-pooling can be represented as two times matrix-matrix multiplications, its back propagation rule is easily derived.

As mentioned in section 2.1, the choice of up sampling operation $\mathcal{U}$ is important. In this paper, $\mathcal{U}$ is set to the inverse process of F-pooling. Specifically, we transform a signal into frequency domain. Then we zero pad the transformed signal to match the resolution of output. Finally, we transform it back to time domain. This process can also be represented by matrix multiplications. For 1D signals, we have:

$$\mathbf{x} = \frac{1}{M}\mathbf{F}_{\mathbf{N}}^* \mathbf{Z}_{\frac{M}{2}} \mathbf{F}_M \mathbf{y} \tag{8}$$

where is $\mathbf{Z}$ is the zero padding operation.

$$\mathbf{Z}_\mu = [\mathbf{x}_{\mathbf{1}:\mu}; \mathbf{0}; \mathbf{x}_{M-\mu+1:M}] \tag{9}$$

The formulations of F-pooling and $\mathcal{U}$ are easily extended to 2D case.

## 2.3 Optimal anti-aliasing down sampling

In this section, we prove that F-pooling is the optimal anti-aliasing down sampling operation from the perspective of reconstruction given $\mathcal{U}$. We focus on 1D case here. Given $\mathcal{U}$, the optimal anti-aliasing down sampling is obtained by solving the following problem:

$$\min_{\mathcal{M}} ||\mathcal{U}\mathcal{M}\mathbf{x} - \mathbf{x}||_2^2, \quad s.t. \quad \mathcal{M} \in \mathcal{A} \tag{10}$$

where $\mathcal{A}$ is a set of all possible anti-aliasing down sampling operations. That is, we hope to find an anti-aliasing down sampling which minimizes the reconstruction error. Based on Nyquist sampling theory, $\mathcal{M}$ must remove high frequencies of signals and this holds for F-pooling. We focus on the optimality of F-pooling. We decompose $\mathbf{x}$ into low frequencies part $\mathbf{x}_l$ and high frequencies part $\mathbf{x}_h$, i.e. $\mathbf{x}_l = \frac{1}{N}\mathbf{F}_{\mathbf{N}}^* \mathbf{D}_{\frac{M}{2}} \mathbf{F}_N \mathbf{x}$ and $\mathbf{x}_h = \mathbf{x} - \mathbf{x}_l$. $\mathbf{D}_\mu$ is a diagonal matrix whose the first $\mu$ and the last $\mu$ diagonal elements equal to 1 while others equal to 0.

$$||\mathcal{U}\mathcal{M}\mathbf{x} - \mathbf{x}||_2^2 = ||\mathcal{U}\mathcal{M}\mathbf{x} - \mathbf{x}_l - \mathbf{x}_h||_2^2 \tag{11}$$

$$= ||\mathcal{U}\mathcal{M}\mathbf{x} - \mathbf{x}_l||_2^2 + ||\mathbf{x}_h||_2^2 + \langle \mathcal{U}\mathcal{M}\mathbf{x}, \mathbf{x}_h \rangle - \langle \mathbf{x}_l, \mathbf{x}_h \rangle \tag{12}$$

$$= ||\mathcal{U}\mathcal{M}\mathbf{x} - \mathbf{x}_l||_2^2 + ||\mathbf{x}_h||_2^2 \tag{13}$$

equation 13 holds because the third term and the forth term of equation 12 equal to 0. $\mathcal{M}$ removes high frequencies and $\mathcal{U}$ doesn't introduce new frequencies. Thus $\mathcal{U}\mathcal{M}\mathbf{x}$ only contains low frequencies. Due to the orthogonality of frequency spectrum, $\langle \mathcal{U}\mathcal{M}\mathbf{x}, \mathbf{x}_h \rangle = 0$. Similarly, $\langle \mathbf{x}_l, \mathbf{x}_h \rangle = 0$. When $\mathcal{M}$ is F-pooling, we have

$$\mathcal{U}\mathcal{M}\mathbf{x} = \frac{1}{MN}\mathbf{F}_{\mathbf{N}}^* \mathbf{Z}_{\frac{M}{2}} \mathbf{F}_M \mathbf{F}_{\mathbf{M}}^* \mathbf{T}_{\frac{M}{2}} \mathbf{F}_N \mathbf{x} \tag{14}$$

$$= \frac{1}{N}\mathbf{F}_{\mathbf{N}}^* \left( \mathbf{Z}_{\frac{M}{2}} \mathbf{T}_{\frac{M}{2}} \right) \mathbf{F}_N \mathbf{x} \tag{15}$$

$$= \frac{1}{N}\mathbf{F}_{\mathbf{N}}^* \mathbf{D}_{\frac{M}{2}} \mathbf{F}_N \mathbf{x} \overset{def}{=} \mathbf{x}_l \tag{16}$$

equation 16 holds due to the definition of operations $\mathbf{T}$, $\mathbf{Z}$, and $\mathbf{D}$. See appendix A for proof. Since the fist term of equation 13 equals to 0, equation 10 is minimized. Thus, we have proved that F-pooling is the optimal anti-aliasing down sampling operation from the perspective of reconstruction.

We choose reconstruction optimality because: 1) we believe reconstruction optimality relates to the final performance, e.g. classification optimality for image classification. If we accept that the feature extracted by previous convolution layers is useful, then it is best to keep it as much as possible for the current pooling layer. Prior works have shown that using self reconstruction loss as an auxiliary is helpful for classification Rasmus et al. (2015); 2) It is difficult to directly define classification optimality for an intermediate layer.

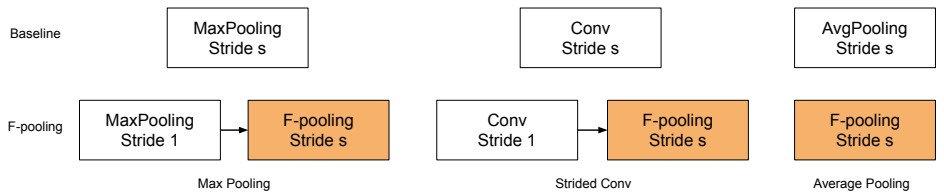

Figure 3: An illustration of how to replace max-pooling, average pooling, and stride convolution with F-pooling. We follow the settings in Zhang (2019).

## 2.4 SHIFT-EQUIVALENT

Based on shift theorem of Fourier transform, we have

$$\mathbf{F}_N Shift_{\triangle h}(\mathbf{x}) = \mathbf{F}_N \mathbf{x} \odot \mathbf{S}_{\triangle h} \tag{17}$$

$$Shift_{\triangle h}(\mathbf{F}_N^* \mathbf{x}) = \mathbf{F}_N^* \mathbf{x} \odot \mathbf{S}_{\triangle h} \tag{18}$$

where $\mathbf{S}_{\triangle h} \in \mathbb{C}^N$ whose $k$th element is $e^{-\frac{2\pi k}{N}\triangle h}$ and $\odot$ is element-wise multiplication. Combining with equation 16, we have

$$\mathcal{U}\mathcal{M}Shift_{\triangle h}(\mathbf{x}) = \frac{1}{N}\mathbf{F}_\mathbf{N}^* \mathbf{D}_{\frac{M}{2}} \mathbf{F}_N Shift_{\triangle h}(\mathbf{x}) \tag{19}$$

$$= \frac{1}{N}\mathbf{F}_\mathbf{N}^* \left( \mathbf{D}_{\frac{M}{2}} \mathbf{F}_N \mathbf{x} \right) \odot \mathbf{S}_{\triangle h} \tag{20}$$

$$= Shift_{\triangle h} \left( \frac{1}{N}\mathbf{F}_\mathbf{N}^* \mathbf{D}_{\frac{M}{2}} \mathbf{F}_N \mathbf{x} \right) \tag{21}$$

$$= Shift_{\triangle h}(\mathcal{U}\mathcal{M}\mathbf{x}) \tag{22}$$

We have proved that F-pooling is shift-equivalent. Note that equation 22 is not hold if other up sampling operations are used, such as linear interpolation.

The proofs in section 2.3 and 2.4 can be easily extended for 2D signals.

## 2.5 TRANSITIVITY OF SHIFT-EQUIVALENT

We study the transitivity of shift-equivalent for F-pooling. Denote $f$ as a shift-equivalent function without down sampling or up sampling. It is straightforward to shown

$$\mathcal{U}\mathcal{M}Shift_{\triangle h}(f(\mathbf{x})) = Shift_{\triangle h}(\mathcal{U}\mathcal{M}f(\mathbf{x})) \tag{23}$$

That is shift-equivalent of F-pooling is transitive for any shift-equivalent function $f$ without down sampling or up sampling. And it is transitive for F-pooling itself. For example, a stack of two F-pooling layers is still a shift-equivalent function.

Unfortunately, this transitivity of F-pooling is not hold for a shift-equivalent function with down sampling. For example, a function of $Fpooling \to conv \to relu \to Fpooling$ is not shift-equivalent. This prevents us to obtain strict shift-equivalent deep CNNs.

## 2.6 PRACTICAL ISSUES

**Dealing with imaginary part:** generally, the output of F-pooling, i.e. $\mathcal{M}\mathbf{x}$ contains both real part and imaginary part. However, for commonly used CNNs, the feature maps must be real. Thus one must ignore the imaginary part of F-pooling. On the other hand, $\mathcal{M}\mathbf{x}$ is treated as complex in the proofs in section 2.3 and 2.4. If we ignore the imaginary part, F-pooling is no longer the optimal anti-aliasing down sampling (but still anti-aliasing) and F-pooling is no longer strict shift-equivalent.

Fortunately, this issue is easy to overcome. When the resolution of down sampled signals is odd, the imaginary part of $\mathcal{M}\mathbf{x}$ is zero. Suppose the resolution is $2\mu + 1$.

$$\hat{\mathbf{x}}_{-\mu}, \ldots, \hat{\mathbf{x}}_{-1}, \hat{\mathbf{x}}_0, \hat{\mathbf{x}}_1 \ldots, \hat{\mathbf{x}}_\mu \tag{24}$$

Table 1: Accuracy and consistency on sub-ImageNet

| accuracy/consistency | densenet-121 | resnet-50 | mobilenet-v2 |
|:---:|:---:|:---:|:---:|
| Origin | 77.47/82.74 | 74.44/80.17 | 73.01/78.08 |
| AA-pooling | 77.14/83.09 | **76.12**/82.26 | 74.03/79.12 |
| F-pooling | **77.56/84.10** | 76.05/**82.63** | **74.72/80.34** |

Due to its symmetry, it only contains real part when transforms back into time domain. If the resolution is $\mu + 2$

$$\hat{\mathbf{x}}_{-\mu}, \dots, \hat{\mathbf{x}}_{-1}, \hat{\mathbf{x}}_0, \hat{\mathbf{x}}_1 \dots, \hat{\mathbf{x}}_\mu, \hat{\mathbf{x}}_{\mu+1} \tag{25}$$

In this case, it contains imaginary part. But we can recover the symmetry and eliminate the imaginary part by setting $\hat{\mathbf{x}}_{\mu+1}$ to zero. We call this trick *odd padding*. We show the effects of *odd padding* in appendix B. A drawback of *odd padding* is that more information is lost during down sampling which may reduces accuracy. We don't use it in this paper.

Compared with equation 24 and equation 25, the imaginary part is introduced by $\hat{\mathbf{x}}_{\mu+1}$. Thus, the error of ignoring the imaginary part is smaller than $||\hat{\mathbf{x}}_{\mu+1}||_2^2$. In practice, we find this effect is small.

**Dealing with zero padding:** F-pooling is designed to be circular shift equivalence. However, zero padding is commonly used in convolution layers. Convolutional layers with zero padding destroy circular shift-equivalence. Thus, one should expect that using circular padding in convolutional layers will greatly increase the shift consistency of F-pooling. In this paper, we evaluate the performance of F-pooling with both zero-padding and circular padding. We suggest one replaces zero padding with circular padding for better shift-equivalence.

### 2.7 IMPLEMENTATION

We implement F-pooling in PyTorch Paszke et al. (2017). Theoretically, it is best to implement F-pooling with fast Fourier transform (FFT) and inverse FFT. Suppose F-pooling down samples an $h_0 \times w_0 \times c$ tensor to a $h_1 \times w_1 \times c$ tensor. Then its time complexity is $c h_0 w_0 \log(w_0 h_0)$. Unfortunately, we find such a implementation in PyTorch is comparatively slow.

Instead, we implement F-pooling via $1 \times 1$ convolutions. As in equation 7, 2D F-pooling can be represented as two times matrix-matrix multiplications along vertical direction and horizontal direction respectively. This is equivalent to $1 \times 1$ convolutions along vertical direction and direction horizontal. Its time complexity is $\mathcal{O}(h_0 w_0 (h_1 + w_1))$ which is higher than the optimal complexity.

F-pooling requires much more computational costs than average pooling or max-pooling. F-pooling is faster than a convolutional layer when the resolution of feature maps is smaller than the number of channels, as the situations of image classifications. Moreover, the number of pooling layers is limited compared with the number of convolutional layers. Thus, introducing F-pooling will not introduce too many computational costs into CNNs.

Zhang (2019) claims that it is important to integrate anti-aliasing pooling into CNNs in a correct way. In this paper, we follow their settings. Specifically, a max-pooling with stride $s$ is replaced with a max-pooling with stride 1 and an F-pooling with stride $s$. A convolution with stride $s$ is replaced with a convolution with stride 1 and an F-pooling with stride $s$. An average pooling with stride $s$ is replaced with an F-pooling with stride $s$. See the illustration in figure 3.

## 3 EXPERIMENTS

### 3.1 1D SIGNALS

We test the shift-equivalent of F-pooling on 1D signals. In figure 1, the original signal is a randomly selected row of a $512 \times 512$ image. We apply max-pooling, average pooling, and F-pooling with stride 4 to those signals. As shown in figure 1, F-pooling is perfectly shift-equivalent. And average pooling is better than max-pooling from shift-equivalent perspective. *odd padding* is used here.

Table 2: Accuracy and consistency on CIFAR-100

| accuracy/consistency | shift argument | densenet-41 | resnet-18 | resnet-34 |
|:---:|:---:|:---:|:---:|:---:|
| Origin | with | 74.90/71.55 | 75.52/70.21 | 76.56/72.21 |
| AA-pooling | with | **75.55**/71.71 | **77.43**/73.08 | 76.95/73.38 |
| F-pooling | with | 75.45/**71.91** | 77.36/**73.43** | **77.68/73.54** |
| Origin | without | 71.81/57.27 | 67.60/45.00 | 68.11/46.56 |
| AA-pooling | without | **73.81**/60.29 | **74.49/58.24** | **74.00/57.72** |
| F-pooling | without | 73.19/**60.51** | 73.93/57.48 | 73.51/56.54 |

## 3.2 Image classification

**CIFAR-100**[1]**:** we test classification of low-resolution $32 \times 32$ color images. This dataset contains 50k images for training and 10k images for test. Images are classified into one of 100 categories.

**sub-ImageNet:** we then test classification of high-resolution color images. Original ImageNet dataset Russakovsky et al. (2015) contains 1.28M training and 50k validation images, classified into one of 1,000 categories. To reduce the computational resources for training, we use a subset of ImageNet (sub-ImageNet) in this work. We randomly select 200 categories from 1,000 categories. For each category, we randomly select 500 images. Thus, we collect 100k images for training. And we select corresponding 10k validation images. All models are trained on a single GPU with batchsize 64 and 100 epochs. We decrease the initial learning rate by a factor 10 every 40 epochs. For other hyper-parameters, we follow PyTorch's official training script.[2]

We train CIFAR-100 using resnet He et al. (2016) and densenet Huang et al. (2017). We train sub-ImageNet using resnet, densenet, and mobilenet-v2 Sandler et al. (2018). Those models are widely used as benchmarks. Max-pooling, average pooling, and stride convolution are covered in those models. We also compare F-pooling with AA-pooling Zhang (2019). For a model trained on sub-ImageNet, its first down sampling operation is kept to reduce computational costs. This setting is also used in Zhang (2019). The results on CIFAR-100 are averaged by 3 runs.

We study not only accuracy but also consistency. We follow the metric of consistency used in Zhang (2019): we check how often the network outputs the same classification, given the same image with two different shifts.

$$\mathbb{E}_{h_0,h_1,w_0,w_1} \mathbf{1} \left\{ \arg\max p(Shift_{h_0,w_0}) = \arg\max p(Shift_{h_1,w_1}) \right\} \tag{26}$$

We only evaluate diagonal shifts in this paper. For CIFAR-100, the number of shifted pixels ranges from -7 to 7. For sub-ImageNet, it ranges from -63 to 63.

**Zero padding:** we keep the padding method of convolutions as zero padding. We measure accuracy and consistency of different pooling methods. Results on sub-ImageNet is shown in table 1 and results on CIFAR-100 are shown in table 2 respectively.

**Circular padding:** as mentioned earlier, circular padding may improve the consistency of F-pooling. Thus we replace the zero padding of convolutions with circular padding. To further evaluate the robustness w.r.t shifts, we use standard deviation (std) of probabilities of the correct label. Std and consistency on sub-ImageNet an CIFAR-100 are shown in table 3 and table 4 respectively.

Based on those results, we conclude that

- F-pooling is significantly and consistently better than original pooling methods in terms of accuracy and shift consistency.

- F-pooling is comparable with AA-pooling in term of accuracy and shift consistency with zero padding. With circular padding, F-pooling is significantly better than AA-pooling.

- Shift consistency of F-pooling is improved more when replacing zero padding with circular padding, especially on CIFAR-100.

---

[1]https://www.cs.toronto.edu/ kriz/cifar.html
[2]https://github.com/pytorch/examples/tree/master/imagenet

Table 3: Std and consistency on sub-ImageNet with circular padding.

| std/consistency | densenet-121 | resnet-50 | mobilenet-v2 |
|---|---|---|---|
| Origin | 0.043/90.44 | 0.051/87.97 | 0.051/87.71 |
| AA-pooling | 0.055/88.21 | 0.056/87.77 | 0.059/86.19 |
| F-pooling | **0.035/91.88** | **0.037/91.01** | **0.041/90.32** |

Table 4: Std and consistency on CIFAR-100 with circular padding without shift argument.

| std/consistency | densenet-41 | resnet-18 | resnet-34 |
|---|---|---|---|
| Origin | 0.111/77.62 | 0.176/59.93 | 0.183/58.94 |
| AA-pooling | 0.166/66.97 | 0.164/65.67 | 0.180/60.53 |
| F-pooling | **0.088/82.99** | **0.073/84.34** | **0.088/80.90** |

## 4 RELATED WORKS

Pooling which reduces the resolution of feature maps is an important part of CNNs. Early CNNs Lecun et al. (1989) use average pooling which is good for anti-aliasing. Later empirical evidence suggests max-pooling Scherer et al. (2010) and strided-convolutions Long et al. (2015) provide better performance. However, small shifts in the input can drastically change the output when stacking multiple max-pooling or strided-convolutions Engstrom et al. (2017); Azulay & Weiss (2018). Other variants such as Graham (2014); He et al. (2015); Lee et al. (2016) (we just list a few of them), focus on extending the functionality of pooling Lee et al. (2016) or making pooling adjusted to variable input size Graham (2014); He et al. (2015).

Recently, Zhang (2019) shows that CNNs will have better shift-equivalent and anti-aliasing properties when low-pass filtering is integrated correctly. However, their method is not strict shift-equivalent and anti-aliasing in theory. Williams & Li (2018) propose Wavelet-pooling. They decompose a signal via wavelet transform and retain the lowest sub-band. This process is repeated until the designed down sampling factor is met. The spirit of Wavelet-pooling is similar to F-pooling. However, they claim that Wavelet-pooling is better than others because it is a global transform instead of a local transform. They neither focus on shift-equivalent nor prove Wavelet-pooling is shift-equivalent or not.

F-pooling is a complex transformation because DFT and IDFT are involved in it. Many works integrate complex transformations or complex values into neural networks. Amin & Murase (2009) study single-layered complex-valued neural networks for real-valued classification problems. Complex numbers represented in polar coordinates are more suitable to deal with rotations. Based on this, Cohen et al. (2018) propose spherical CNNs which are rotation-equivalent to deal with signals projected from spherical surface. Trabelsi et al. (2018) propose general deep complex CNNs. They adjust batch normalization and non-linear activations for complex CNNs. F-pooling can used in their method without worry about the imaginary part. F-pooling utilizes shift theorem of DFT and achieves shift-equivalent. This is a new success for the combination of complex transformations and neural networks.

## 5 CONCLUSIONS

In this paper, we have proposed frequency pooling (F-pooling) for CNNs. F-pooling reduces the dimension of signals in frequency domain. We have defined shift-equivalent of functions which contain down sampling operations by introducing an up sampling operation. Under this definition, we have proved that F-pooling is the optimal anti-aliasing down sampling operation and F-pooling is shift-equivalent. We have integrated F-pooling into modern CNNs. We have verified that F-pooling remarkably increases accuracy and robustness w.r.t shifts of modern CNNs. We believe that F-pooling plays a more important role in applications where shift-equivalent is more serious, such as object detection and semantic segmentation.

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

## A    PROOF OF OPERATION

We prove $\mathbf{D}_{\frac{M}{2}} = \mathbf{Z}_{\frac{M}{2}}\mathbf{T}_{\frac{M}{2}}$. By definition, $\mathbf{T}$ can be represented as an $M \times N$ matrix and $\mathbf{Z}$ can be represented as a $N \times M$ matrix.

$$\mathbf{T}_{\frac{M}{2}} = \begin{bmatrix} \mathbf{I}_{\frac{M}{2}} & \mathbf{0}_{\frac{M}{2}} & \mathbf{0}_{\frac{N}{2}-M} & \mathbf{0}_{\frac{N}{2}-M} & \mathbf{0}_{\frac{M}{2}} & \mathbf{0}_{\frac{M}{2}} \\ \mathbf{0}_{\frac{M}{2}} & \mathbf{0}_{\frac{M}{2}} & \mathbf{0}_{\frac{N}{2}-M} & \mathbf{0}_{\frac{N}{2}-M} & \mathbf{0}_{\frac{M}{2}} & \mathbf{I}_{\frac{M}{2}} \end{bmatrix} \tag{27}$$

where $\mathbf{I}_M$ is an $M \times M$ identity matrix and $\mathbf{0}_M$ is an $M \times M$ zero matrix. $\mathbf{Z}_{\frac{M}{2}}$ is equal to the transpose of $\mathbf{T}_{\frac{M}{2}}$ by definition. Then

$$\mathbf{Z}_{\frac{M}{2}}\mathbf{T}_{\frac{M}{2}} = \begin{bmatrix} \mathbf{I}_{\frac{M}{2}} & \cdot & \cdot \\ \cdot & \mathbf{0}_{N-M} & \cdot \\ \cdot & \cdot & \mathbf{I}_{\frac{M}{2}} \end{bmatrix} \tag{28}$$

which is the same as $\mathbf{D}_{\frac{M}{2}}$ by definition.

## B    EFFECT OF *odd padding*

## C    LOSS CURVES ON SUB-IMAGENET

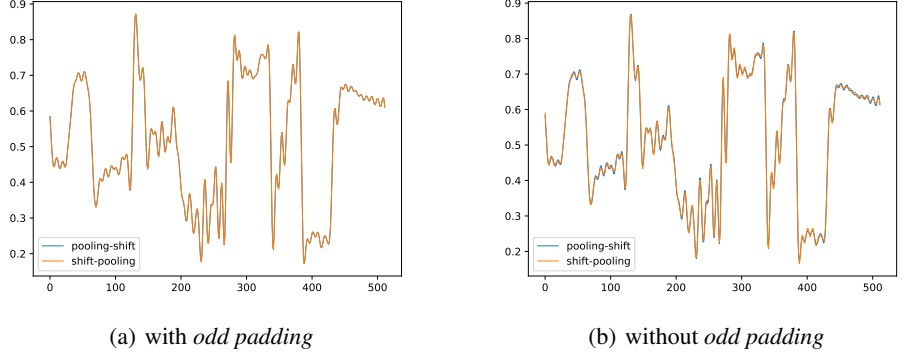

(a) with *odd padding*                     (b) without *odd padding*

Figure 4: With *odd padding*, F-pooling is strict shift-equivalent. Without it, F-pooling is not strict shift-equivalent. But the error is acceptable. Best viewed on screen.

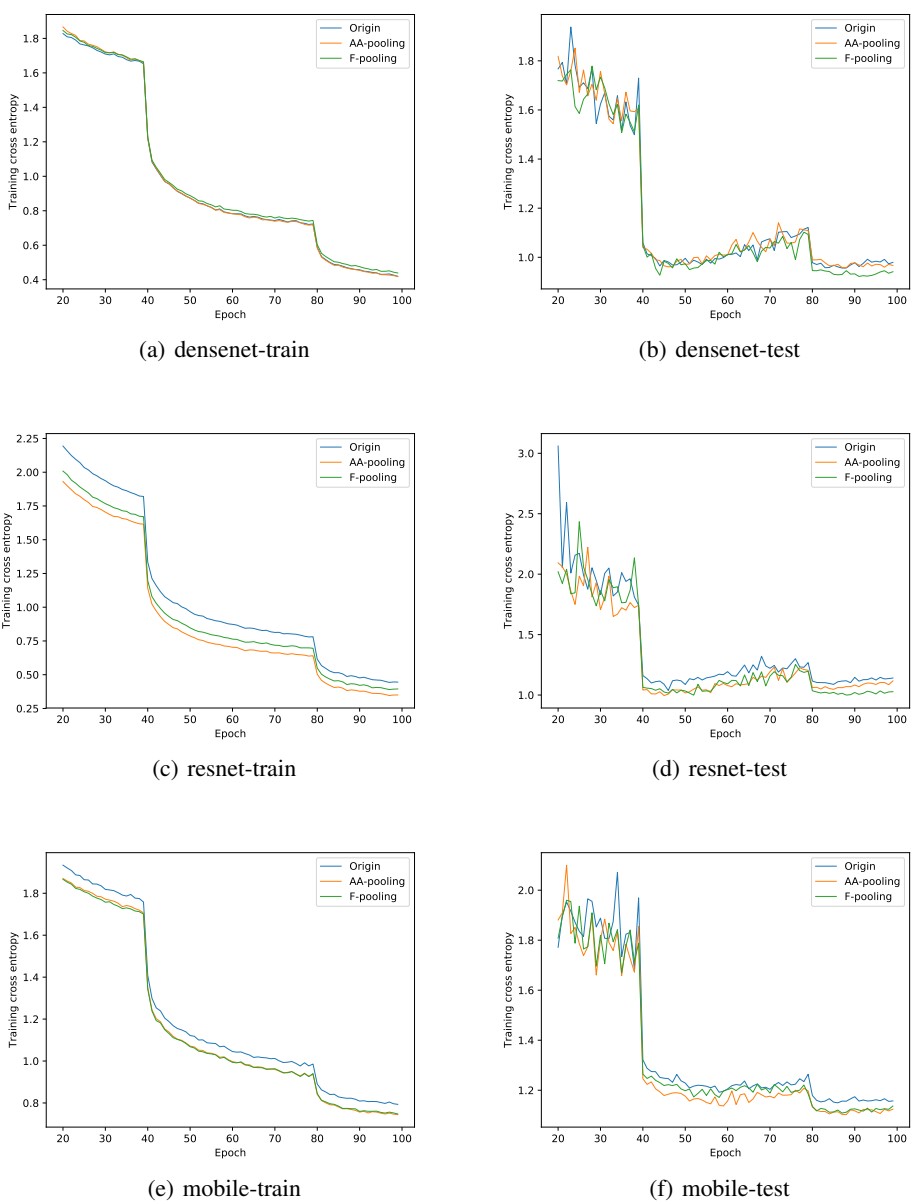

(a) densenet-train

(b) densenet-test

(c) resnet-train

(d) resnet-test

(e) mobile-train

(f) mobile-test

Figure 5: Loss curves on sub-ImageNet. Best viewed on screen.

