# OpenReview forum: "Frequency Pooling: Shift-Equivalent and Anti-Aliasing Down Sampling"
_ICLR.cc/2020/Conference — Reject_

### Official Review · AnonReviewer4 · 2019-10-17
**Official Blind Review #4**

**Rating:** 1

**Review:**

This paper proposed "F pooling" for Frequency Pooling, which is a pooling operation satisfying shift equivalence and anti-aliasing properties. The method is very simple: first, transform the input 1D/2D signal into the spectrum domain based on discrete Fourier transform (DFT), then cut the high-frequencies, then transform back to the time domain using the inverse DFT. The method can be implemented using FFT and auto differentiation frameworks. The method is tested on Resnet/Desnet on CIFAR-100 and subsets of ImageNet, showing better performance than the original models.

The reviewer votes for rejection as the method has limited novelty. Spectrum pooling has been used in the community of computer vision and machine learning. Taking a random example (there are others by simple searching), in the ECCV paper "DFT-based Transformation Invariant Pooling Layer for Visual Classification, Ryu et al., 2018" The DFT magnitude pooling is almost the same as the authors' propositions, where the "Fourier coefficients are cropped by cutting off high-frequency components".

The reviewer encourages the authors to make further new developments and have a more comprehensive literature review. But in the current form, the paper has less value to be published in ICLR.

**Experience Assessment:**

I have read many papers in this area.

**Review Assessment: Checking Correctness Of Derivations And Theory:**

I carefully checked the derivations and theory.

**Review Assessment: Checking Correctness Of Experiments:**

I assessed the sensibility of the experiments.

**Review Assessment: Thoroughness In Paper Reading:**

I read the paper thoroughly.

---

> ### Author Response · Authors · 2019-11-08
> **Response to Review #4**
>
> Thanks for your comments.
> We are sorry to say that we miss some previous works, especially the ECCV one. And we will provide more literature review in our updated paper. We admit that the computation process of F-pooling and the ECCV method is the same.
>
> However, we defend the novelty of our work. The values of our work are not how the output of F-pooling is computed. Instead, the values are the strict definition of shift-equivalence and the theoretical properties of F-pooling. In previous works, they even don’t give an operable definition shift-equivalence when down sampling involved.
>
> Please refer to our general response for more of F-pooling’s values.
>
> Moreover, we discuss some practical problems of F-pooling. Such as how to deal with the imaginary part and the zero-padding of convolutions. With suitable settings, the shift consistency of F-pooling is much better.

---

> > ### Comment · AnonReviewer4 · 2019-11-15
> > **Acknowledging rebuttal**
> >
> > I have read the authors' rebuttal.
> >
> > I summarized the authors' response as follows:
> > (1) the proposed computation is not novel;
> > (2) the novelty lies in:
> > (2a) a strict definition of shift-equivalence;
> > (2b) theoretical properties of F-pooling.
> >
> > For (2a), the author's definition is an adaptation of (Zhang 2019) plus the pooling setting and is, therefore, a special case of the shift-equivalence discussed by Zhang (2019).
> >
> > For (2b), I could not find where the theoretical results are presented, besides the derivation to show that F-pooling satisfies the shift-equivalent property. As far as I understand, this is a technical contribution (rather than theoretical), where the usual storyline is to propose a new method and show its empirical results.
> >
> > Overall, the novelty is indeed limited. As the authors are still updating their experimental results, I couldn't have enough reasons to update my score.

---

> > > ### Author Response · Authors · 2019-11-15
> > > **rebuttal**
> > >
> > > 1. Our definition is not the special case of (Zhang 2019). In fact, their definition is not correct for functions where down sampling involved. In our definition, an up sampling operator is used. This operator plays an important role in our proofs.
> > >
> > > 2. Our theoretical results are presented in section 2.2, 2.3 and 2.4. In section 2.3, we prove that F-pooling is the optimal anti-aliasing down sampling. We suggest you read that part carefully.
> > >
> > > These results can't be developed based on the previous definition of shift-equivalent. These results are derived based on our strict definition.
> > >
> > > In summary, we provide a mathematical treatment of this problem and develop some results. Thus we believe we have theoretical contributions.
> > >
> > > Moreover, we discuss some practical issues such as how to deal with the imaginary part.
> > >
> > > 3. We apologize for our mistake when calculating the shift consistency. The updated results are not new, but to fix our code mistake.

---

### Official Review · AnonReviewer2 · 2019-10-29
**Official Blind Review #2**

**Rating:** 6

**Review:**

This paper proposed a new pooling method (Frequency pooling) which is strict shift equivalent and anti-aliasing in theory. The authors first derived the theory of F-pooling to be optimal anti-aliasing down sampling and is shift-equivalent in sec 2, and then demonstrated the experimental results of 1D signals and image classification tasks.

The experimental results are actually less impressive than what are claimed in contribution and conclusion. The authors stated that "F-pooling remarkably increases accuracy and robustness w.r.t. shifts of moderns CNNs"; however, in Table 1-3, the winning margin of accuracy is actually quite small (<2%), and the consistency (<3.5% compared to the second best baseline except resnet-18 on CIFAR 100 has large improvement ~7-8%).

Questions:
1. For the experiment of 1D signal on sine wave, the AA-pooling and F-pooling give the same result?
2. Compared to AA-pooling, it seems that F-pooling has a better theoretical guarantee (i.e. the optimal anti-aliasing down sampling operation given U). But other than this, the empirical performance seem not showing particular advantage over AA-pooling. Are there any other advantages for F-pooling s.t. people might want to use it as opposed to AA-pooling?
3. What are the limitations of the F-pooling? It is good to me that the authors discuss one limitation on the imaginary part of output and I would like to hear more on other potential limitations for this method.
- also, if the authors can explain more on sec 2.5 it will be helpful. If we simply ignore the imaginary part, although the theory is not applicable, but what would the empirical performance be?



**Experience Assessment:**

I do not know much about this area.

**Review Assessment: Checking Correctness Of Derivations And Theory:**

I assessed the sensibility of the derivations and theory.

**Review Assessment: Checking Correctness Of Experiments:**

I assessed the sensibility of the experiments.

**Review Assessment: Thoroughness In Paper Reading:**

I read the paper at least twice and used my best judgement in assessing the paper.

---

> ### Author Response · Authors · 2019-11-08
> **Response to Review #2**
>
> Thanks for your comments.
> To our knowledge, close to 2% improvement of accuracy is not small in CIFAR100. Because we only change pooling layers while keeping others exactly the same.
>
> Now, we respond to your questions one by one:
>
> 1. The results of AA-pooling and F-pooling are not the same. In Fig. 1, we show the results of average-pooling and F-pooling. If you carefully look at the corner of curves, you can find the differences. Without convolution, AA-pooling is similar to average-pooling (both of them are low-pass filters but with sightly different kernels. So AA-pooling gives different results for sine waves.
>
> 2. We believe F-pooling plays a more important rule in applications where shift-equivalent is serious, such as object detection and object tracking. Because we need to predict the location or shifts of an image object. Moreover, F-pooling may be better for complex-valued CNNs, such as [1].
>
> 3. The limitation of imaginary part is easy to overcome: set the resolution of F-pooling’s output to an odd number or padding it to an odd number when the resolution is an even number. In this way, the imaginary part is zero. Moreover, the word shift in this paper means circular shift. So it is better to use circular padding in convolutional layers. However, we find circular padding slower the training speed in PyTorch. If we use zero paddings as in most situations, the beneficial of F-pooling is reduced. Our current experiments use zero paddings. See our general response for what happens when we replace zero paddings with circular padding.
>
> 4. In all experiments of our current paper, the imaginary part is already ignored.
>
> We can’t directly measure how the imaginary part affects the performance unless we use complex-valued CNNs. Ignoring this part will destroy the reconstruction optimality, but the effect is small. Suppose the output size of F-pooling is 2N+1. We first transform a signal into frequency domain and keep 2N+1 components with the lowest frequencies: f(-N), … , f(0), … ,f(N). Then we transform it back into time domain. In this case, the imaginary part in time domain is zero because of symmetry. Now, suppose the output size is 2N+2: f(-N), … , f(0), … f(N), f(N+1). In this case, the imaginary part is not zero. However, if we set f(N+1) to 0, it imaginary part becomes zero again. Thus, the error of ignoring imaginary part is not larger than ||f(N+1)||. Fig.4 shows an example of odd and even output size of F-pooling.
>
> [1] Deep complex networks, ICLR2018

---

### Official Review · AnonReviewer1 · 2019-11-01
**Official Blind Review #1**

**Rating:** 3

**Review:**

This paper researches the pooling operation, which is an important component in convolutional neural networks (CNN) for image classification. Taking the perspective from signal processing, this paper proposes a pooling operation called frequency pooling (F-pooling). The key motivation is to make the pooling operation shift-equivalent and anti-aliasing. This paper gives an improved definition on shift-equivalent functions and shows that the proposed F-pooling is optimal in the sense of reconstructing the orignal signal. The F-pooling is then implemented with matrix multiplications and tested with recent convolutional neural networks for image classifiation on CIFAR-100 and a subset of ImageNet dataset.

It is interesting to take the perspective from signal processing to give pooling operation in CNN a formal treatment. As indicated in the recent literature, enforcing shift-invariance does help to improve the performance of a CNN on classification accuracy and the robustness with respect to image shift. At the same time, this work can be further enhanced at the following aspects:
1. This work can make it clearer in principle how anti-aliasing contributes to improving the classification performance and robustness. This will help to make this paper more self-contained.
2. When showing the optimality of F-pooling in Section 2.3, the criterion is to reconstruct the original signal x. Considering that the ultimate goal is classification, the information to be maximally preserved through each operation through the layers shall be the information that relates to the class label y. In light of this, some justification and explanation shall be provided for using this criterion for optimality.
3. The experimental study is weak. Experiments could be conducted on more benchmark datasets with more CNN architectures to convincingly show the effectiveness of the proposed F-pooling. Also, from the three Tables in the experimental part, the improvement of F-pooling over AA-pooling (developed by the main reference of this work) does not seem to be significant or consistent. For example, in Table 2, the F-pooling only wins at either accuracy (marginally) or consistency, but not both. In Table 3, the F-pooling consistently shows inferior classification performance, although obtaining slightly higher consistency. This makes the advantage of F-pooling over the existing AA-pooling unclear.

**Experience Assessment:**

I have read many papers in this area.

**Review Assessment: Checking Correctness Of Derivations And Theory:**

I assessed the sensibility of the derivations and theory.

**Review Assessment: Checking Correctness Of Experiments:**

I assessed the sensibility of the experiments.

**Review Assessment: Thoroughness In Paper Reading:**

I read the paper at least twice and used my best judgement in assessing the paper.

---

> ### Author Response · Authors · 2019-11-08
> **Response to Review #1**
>
> We think your suggestions are very meaningful. We respond to them one by one:
>
> 1. We will explain anti-aliasing in our updated paper. Roughly, anti-aliasing is helpful for signal reconstruction. However, we can’t provide a strict treatment of how anti-aliasing relates to classification. But we have intuitions: first, we believe reconstruction relates to classification (see our next response); second, frequency components are orthogonal. Aliasing means different components are mixed again. This may mislead the next layers for processing.
>
> 2. To our knowledge, researchers haven’t fully understood the whole process of image classification until now. Thus, we can’t provide a strict treatment of how reconstruction optimality relates to classification optimality. But we have intuitions and empirical evidence of their relation: convolution layers are used to transform a signal which makes it easier to be classified. So if we accept that the feature extracted by previous convolution layers is useful, then it is best to keep it as much as possible for the current pooling layer. In this way, it is reasonable to assume that reconstruction optimality is consistent with classification optimality. On the other hand, it is difficult to directly define classification optimality for an intermediate layer. Moreover, several works, such as [1] have shown that using self reconstruction loss as an auxiliary is helpful for classification.
>
> 3.  Please refer to our general response. With suitable settings, the shift consistency of F-pooling is much better.
>
> [1] Semi-Supervised Learning with Ladder Networks, NIPS2015

---

### Author Response · Authors · 2019-11-08
**General response to all reviewers:**

Thank you for your comments and suggestions. We are sorry that our paper is written in a hurry. We accept that some contents are not explained well and the experiments are weak in the current version of this paper.

However, we defend the value and novelty of our work in the following aspects:

1. We believe shift-equivalence is very important for CNNs. Our work aims to recovery this property of CNNs which are destroyed by down sampling in a principle way. We provide theoretical guarantees of optimal anti-aliasing and shift equivalence.

2. Although everyone talks about shift-equivalence, we find there is no strict definition of it for CNNs when down sampling is involved. We have shown that a corresponding up sampling operation must be involved in the strict definition. We also have shown that this up sampling operation plays an important role when we prove the properties of F-pooling. We believe a formal definition and a mathematical treatment have great value for academic research.

3. Some reviewers mention that F-pooling is not consistently better than AA-pooling. But we never claim that F-pooling must beat AA-pooling. We choose AA-pooling just to provide a performance reference of the recently proposed method in this topic. Experiments in Tab 1-3 are consistently better than the baseline. We think this is enough to show the effectiveness of F-pooling.

4. We choose image classifications to evaluate F-pooling just because it is commonly used and is easy to implement. There are some computer vision applications where shift-equivalence plays a more important role, such as object detection and object tracking. In those applications, F-pooling may be more valuable.

************************************************************************************************
We apologize again because we make a terrible mistake when we test shift consistency. The consistency results in all table are incorrect. We have updated the correct values in our new pdf files. The relative order is similar as before.

One of the most important reasons why the consistency of F-pooling is not as good as we expect is that: F-pooling is designed to be circular shift equivalence. However, convolutional layers with zero paddings destroy circular shift equivalence. Thus, one should expect that using circular padding in convolutional layers will greatly increase the shift consistency of F-pooling.

When we use circular padding, the shift consistency of F-pooling becomes much better. For ResNet18 on CIFAR100 with circular padding without shift argument, we have the following attractive results:
Baseline:          60.47
AA-pooling:      65.14
F-pooling:         84.04

************************************************************************************************
We will revise our paper based on your comments and update our paper during rebuttal. We plan to add more experiments to show the effect of F-pooling better. We also plan to submit our code during rebuttal.

---

### Author Response · Authors · 2019-11-15
**To all reviewers: our paper is updated.**

We have updated our paper based on your suggestions and comments.

We study how zero padding and circular padding of convolutions affect the results. F-pooling is beneficial from circular padding more than AA-pooling and baseline. The results are shown in table 3 and table 4.

We believe we are dealing with a fundamental problem of CNNs which is ignored by the community: pooling destroys the shift-equivalent of CNNs. We provide a strict definition of shift-equivalent when down sampling involved and develop some results based on it.

We believe that F-pooling plays a more important role in applications where shift-equivalent is more serious, such as object detection and semantic segmentation.

*****************************************
Our anonymous code is released.

---

### Decision · Program_Chairs · 2019-12-19

**Decision:**

Reject

**Comment:**

This submission has been assessed by three reviewers and scored 3/6/1. The reviewers also have not increased their scores after the rebuttal. Two reviewers pointed to poor experimental results that do not fully support what is claimed in contributions and conclusions. Theoretical support for the reconstruction criterion was considered weak. Finally, the paer is pointed to be a special case of (Zhang 2019). While the paper has some merits, all reviewers had a large number of unresolved criticism. Thus, this paper cannot be accepted by ICLR2020.